# Characteristics and Quality Analysis of Radio Frequency-Hot Air Combined Segmented Drying of Wolfberry (*Lycium barbarum*)

**DOI:** 10.3390/foods11111645

**Published:** 2022-06-02

**Authors:** Yanrui Xu, Zepeng Zang, Qian Zhang, Tongxun Wang, Jianwei Shang, Xiaopeng Huang, Fangxin Wan

**Affiliations:** College of Mechanical and Electrical Engineering, Gansu Agricultural University, Lanzhou 730070, China; yangh@st.gsau.edu.cn (Y.X.); zangzp@st.gsau.edu.cn (Z.Z.); zhangq@st.gsau.edu.cn (Q.Z.); wangtx@st.gsau.edu.cn (T.W.); shangjw@st.gsau.edu.cn (J.S.); wanfx@gsau.edu.cn (F.W.)

**Keywords:** wolfberry, radio frequency, hot air, combined segmented drying, drying characteristics, quality analysis

## Abstract

To overcome the problems of a long conventional drying time, low energy efficiency, and poor product quality, a segmented drying approach was developed for fresh wolfberry (*Lycium barbarum*) using a radio frequency (RF)-hot air drying process, which was investigated under different parameters of plate spacing (80, 90, 100 mm), vacuum degree (0.015, 0.025, 0.035 Mpa), and hot air temperature (50, 55, 60 °C). Analysis of the wolfberry’s drying characteristics, comprehensive quality, and microstructure indicated that: combined drying was faster and less time-consuming than natural drying or hot air drying, and components such as polysaccharides, ascorbic acid, and betaine in wolfberries were effectively retained. Based on the acceptable drying rate, stable temperature application, and avoidance of arcing effects, the optimal combined segmented drying parameters were determined to be as follows: a plate spacing of 90 mm, vacuum degree of 0.025 MPa, and air temperature of 55 °C. For the dried wolfberries under these conditions, the total drying time was 17 h and the berries had an improved comprehensive quality, the content of total soluble sugars was 0.62 g/g, total phenol was 10.01 mg/g, total flavonoids was 2.60 mg/g, V_C_ was 3.18 mg/100 g, betaine was 3.48%, oxidation resistance represented by an inhibition rate was 66.14%, color was better, and rehydration rate was 48.56%. The microstructure was more regular because of the special dielectric heating characteristics of RF vacuuming. Despite the differing drying characteristics of individual materials, the overall RF-hot air combined drying process was found to achieve high-quality dehydration of wolfberries.

## 1. Introduction

Wolfberry (*Lycium barbarum*) is a perennial deciduous shrub in the family Solanaceae that is mainly planted in northern China and produces orange-red ellipsoid berries [1]. The length of a fresh wolfberry is 1–2 cm and the annual production can reach 0.26 million tons; most of them are consumed and applicated after being dried [2,3]. Pharmacological studies have shown that polysaccharides, total phenols, total flavonoids, and other functional compounds in drying-processed wolfberries are immune-supportive, anti-tumorigenic, and have anti-aging effects [4,5]. The moisture content of ripe, fresh wolfberry fruit can reach more than 80%, and the fruit has a high sugar content and is sensitive to microbial deterioration, causing it to rot easily during harvest and storage [6]. Therefore, harvested fresh wolfberry must be dehydrated to a safe moisture content in a timely manner to facilitate subsequent transportation, storage, and processing. In addition to traditional natural drying, hot air, microwave vacuum, far-infrared, and vacuum freeze drying technologies have been applied in the drying of wolfberry. However, the energy utilization efficiency and product quality of these drying methods are difficult to control: a longer drying time with hot air causes serious shrinkage and nutrient loss in dried wolfberries [7]; the energy utilization rate of microwave vacuum drying is higher, but it easily causes surface expansion and rupture, seeds to fall out, and sugar overflow [8]; after vacuum far-infrared drying, the surface is prone to crust-hardening and forms small bubbles which affects the wolfberry quality [9]; and vacuum freeze drying technology can effectively retain the bioactive components of dried wolfberries, but typically has a high energy consumption, low yield, and requires expensive equipment [10]. Therefore, appropriate post-harvest drying is an important step to increase the total quality and effective component retention rate of the wolfberry.

In recent years, radio frequency (RF) heating has emerged as a new environmental-protection drying technology that imposes friction by the oscillation of charged ions and rotation of polar molecules to produce heat inside the material [11]. RF has a better penetration ability, can affect the overall rapid heating of materials, and is widely used in the treatment of agricultural products after harvest [11,12]. Relevant research shows that when nut products such as walnuts [13,14] and hazelnuts [15,16] are dried by a RF vacuum, the internal kernel temperature is higher than the shell temperature, which can effectively remove the kernel moisture, shorten the drying time, and improve the quality of nuts. When applied to the drying of food crops such as soybeans [17,18,19], corn [20], and rice [21], seed moisture can be quickly dried to ensure for stable storage and that pests can be effectively controlled. However, RF drying is limited for materials with a high moisture content that are easy to rupture and experience shrinkage of the outer skin. For example, selecting a certain electrode gap and vacuum degree can achieve higher quality dehydration of kiwifruit [22,23], apple [24], and mango fruit slices [25], as their nutrient retention rate and rehydration rate are comparatively high, but they need to be sliced with an appropriate thickness (6–8 mm) in the early stage of drying, thus preventing a low equilibrium temperature leading to excessive preheating and drying times.

Although the drying rate by RF heating alone is rapid, thermal deviation and corner effects can easily cause the decline in product quality due to the glow discharge phenomenon caused by excessive or uneven heating [23]. Hot air-assisted RF heating can effectively improve the uniformity of heating, greatly improve the drying rate, and reduce the energy consumption, as has been demonstrated when applied to insecticidal and drying fields [11,12]. It was reported that the total phenol content, shrinkage rate, and color change of bananas after RF treatment at the early stage of hot air drying showed an upward trend [26,27], and the rehydration rate and hardness of samples decreased. Hot air-assisted RF drying was also applied to the drying of apricots [28,29], and the experimental results showed that the drying process had significant effects on the color, antioxidant capacity, and β-carotene content of preserved apricots. In addition, the drying time was significantly reduced. When hot air-assisted RF technology was used in the final stage of pre-dried carrot slices [30], the heating uniformity and drying rate were good, and the drying time was shortened by more than 50% compared with conventional hot air drying; furthermore, the color, hardness, chewiness, and ascorbic acid content (V_C_) retention rate were improved over conventional drying. In summary, RF-hot air combined drying can be widely used as a high-quality dehydration process in the drying of some agricultural products after harvest.

In recent years, radio frequency drying has been used in the drying process of many materials, but its application in wolfberry drying and an analysis of the drying rate and quality have not been reported. In this study, fresh wolfberries were dried by RF-hot air combined segmented drying technology in an attempt to establish the optimal combined drying process parameters. The effects of RF-hot air combined drying technology were assessed with respect to the drying rate, product quality, and microstructure compared with natural drying and single hot air drying to provide technical guidance for postharvest processing.

## 2. Materials and Methods

### 2.1. Experimental Materials

The variety of wolfberry used in this experiment was the fresh fruit of Ningqi No. 1 planted in a Zhongning *L. barbarum* plantation in Ningxia, China, and was purchased and stored in a temperature-controlled cabinet at (2 ± 1 °C) prior to testing [31,32].

### 2.2. Instruments and Equipment

The GJS-3-27-JY RF vacuum dryer was used in this study (Hebei Huashi Jiyuan High Frequency Equipment Co., Ltd., Langfang, China). As shown in Figure 1, the equipment mainly included a control panel and negative pressure tank (diameter 800 mm, length 800 mm), an RF device with upper and lower plates (600 × 400 mm), an elevating device, a water tank, an optical fiber temperature sensor, a weighing sensor, and a pressure sensor. The system had the following characteristics: The RF device was located in the negative pressure tank. The upper and lower plates transmitted a high-frequency current. The elevating device comprised a ball screw and servo motor, which was negatively sealed with a rubber O-ring in the pressure tank after connecting with the plate. After setting the parameters on the control panel, the solid-state relay of the servo motor and vacuum pump was controlled by an RS-485 bus to adjust the plate spacing and the vacuum degree in the negative pressure tank. In addition, the temperature, weight, and pressure of the material in the negative pressure tank were measured and transmitted to the control panel in real-time through four fiber-optic probe sensors installed in the negative pressure tank, with a weight sensor fixed below the lower plate (≤5 kg) and a pressure sensor installed on the inner wall of the negative pressure tank (≥−0.09 MPa). The negative pressure tank condensed the water vapor generated during the drying process, which was periodically collected through the separation valve into the tank to maintain a constant drying chamber humidity.

Other instruments and equipment: a YQ101-0A-4A electrically heated drum dryer (voltage 220 V/50 Hz, power 1.2 kW, wind speed 3 m/s; Beijing Yuqin Tengda Pharmaceutical Equipment Co., Ltd., Beijing, China); an AUY220 electronic balance (Shimadzu, Kyoto, Japan); a CS-210 precision colorimeter (Zhengzhou North-South Instrument Co., Ltd., Zhengzhou, China); an HKSF-2 Quick Moisture Meter (Moisture Resolution 0.01%, Precision 0.1%; Wuxi Huake Instrument Co., Ltd., Wuxi, China); a TS-200B table-type constant temperature rocker (Shanghai Jinwen Equipment Co., Ltd., Shanghai, China); a CL5 large capacity low-speed centrifuge (Changsha High-tech Industrial Development Zone Xiangyi Centrifuge Instrument Co., Ltd., Changsha, China); a V-5100 spectrophotometer (Shanghai Yuanshi Instrument Co., Ltd., Shanghai, China); a KQ-300VDE three-frequency CNC ultrasonic cleaner (Kunshan Ultrasonic Instrument Co., Ltd., Kunshan, China); an Agilent 1100 high-performance liquid chromatograph (Agilent, Palo Alto, Santa Clara, CA, USA); and a Sigma-300 scanning electron microscope (Zeiss, Oberkochen, Germany).

### 2.3. Experimental Methods

Wolfberries were removed from the temperature-controlled cabinet 1 h before the experiment and allowed to equilibrate to room temperature (22 °C ± 1 °C). Samples with no surface damage, high moisture content (79 ± 2%), and similar shape and size (1.5–2.0 cm) were selected. Each experiment selected 120 ± 0.5 g of whole fresh wolfberries. The wolfberry samples were immersed in 2% Na_2_CO_3_ solution for 5 min, as the purpose of the alkali pretreatment was to dissolve the surface wax layer. Then, the samples were drained of surface moisture, evenly spread on the polypropylene plate (500 × 300 mm), and placed into the RF vacuum drying equipment drying chamber for the first stage of drying. In order to avoid the corner effect and glow discharge phenomenon of RF drying, the pre-experiment was carried out first. According to the pre-experiment results, different parameters of plate spacing (80, 90, 100 ± 0.5 mm), vacuum degree (0.015, 0.025, 0.035 ± 0.002 Mpa), and hot air temperature (50, 55, 60 ± 0.5 °C) were selected.

The pre-experimental results showed that the surface of a wolfberry is hardened and even burned when the moisture ratio was reduced to 0.4–0.5 g/g. When heated by radio frequency, the epidermis and the interior of the wolfberry were heated simultaneously. Although the moisture content of the epidermis is relatively small, with the extension of heating time, the free water in the epidermis of the wolfberry evaporates faster than that in the interior, the moisture content decreases, the free water transforms to bound water, and the molecular viscosity increases so that it becomes difficult for the water molecules to move and diffuse. Therefore, the moisture ratio of 0.4–0.5 g/g was used as the joint drying node and samples were weighed every 60 min. When the moisture node was reached, the samples were transferred to the hot air drying equipment for the second stage of drying to evaporate the internal moisture of the wolfberries. The total operation time was less than 1 min and the temperature loss was less than 5 °C during the transfer process. The weighing interval was the same as the first stage until the constant weight was reached. The moisture ratio 0.10–0.12 g/g was selected for the drying end node and the corresponding moisture content was 10 ± 0.5%. The wolfberries dried by natural drying and single hot air drying were regarded as control groups, and the experiment was repeated three times under each group of parameters.

### 2.4. Calculation of Drying Parameters

#### 2.4.1. Determination of Moisture Content

A total of 10 g of fresh wolfberries was weighed and spread on the material plate of the HKSF-2 rapid moisture meter; the measured average initial moisture content was 79 ± 2%.

#### 2.4.2. Calculation of Dry Basis Moisture Content

The moisture content of the dried base in the drying process of the wolfberries was calculated as follows:(1)$$M_{t}=\frac{m_{t}-m_{g}}{m_{g}},$$
where *M_t_* is the moisture content of the dried base at time *t* of medlar; *m_t_* is the mass of wolfberries at time *t* (g); and *m_g_* is the dry matter mass of wolfberries (g).

#### 2.4.3. Calculation of Moisture Ratio

The moisture ratio of the wolfberries during drying was calculated as follows:(2)$$MR=\frac{M_{t}-M_{e}}{M_{0}-M_{e}},$$
where *MR* is the moisture ratio of wolfberries; and *M*_0_, *M_e_*, and *M_t_*, are the initial dry basis moisture content, the dry basis moisture content at drying to equilibrium, and the dry basis moisture content of materials at time t, respectively. Because *M_e_* is far less than *M*_0_ and *M_t_*, it can be approximated to 0; thus, the formula of the moisture ratio of wolfberries at different drying times can be simplified as Equation (3) [33]:(3)$$MR=\frac{M_{t}}{M_{0}},$$

The drying rate of wolfberries during drying is calculated as follows [34]:(4)$$V_{R}=\frac{M_{t_{1}}-M_{t_{2}}}{t_{2}-t_{1}},$$
where *V_R_* is the drying rate of wolfberries; *t*_1_ and *t*_2_ can be used to indicate any drying time; and *M_t_*_1_ and *M_t_*_2_ are the dry base moisture content of wolfberries at *t*_1_ and *t*_2_, respectively.

### 2.5. Determination of Quality Indicators

#### 2.5.1. Determination of Color

The three dried wolfberry samples under each group of different drying parameters were randomly selected for colorimetric analysis measured by a CIELab system, the reference of the illuminant was D65, SCI, and the degree of the observer was 10°. Three whole samples in each group were measured and color calculated as follows [22,23,24,25]:(5)$$\Delta E=\sqrt{\;(L-L^{*}){\;}^{2}+(a-a^{*}){\;}^{2}+(b-b^{*}){\;}^{2}},$$
where Δ*E* denotes total chromatic aberration; *L*, *a*, and *b* represent the brightness value, red and green value, and yellow and blue value of fresh wolfberry samples, respectively; and *L**, *a**, and *b** denote the brightness value, red and green value, and yellow and blue value of dried wolfberry products, respectively.

#### 2.5.2. Determination of Rehydration Rate

According to the special method of making edible dried wolfberries, approximately 5 g of dried wolfberries was weighed and placed in a small beaker; then, 60 mL of hot water (80 °C) was added and placed in a constant temperature water bath at 80 °C [35]. Surface moisture was removed by vacuum filtration on clean filter paper after 20 min and quality was assessed after 30 min in shaded conditions, calculated as follows [35,36,37]:(6)$$N=\frac{m_{2}\cdot \left(1-w_{0}\right)\;}{m_{1}\cdot \left(1-w_{1}\right)\;}\times 100\%,$$
where *m*_1_ is the quality of the dried wolfberries (g); *m*_2_ is wolfberry quality after rehydration (g); *w*_0_ is the initial moisture content of the fresh wolfberries (%); and *w*_1_ is the moisture content of the wet basis of the dried wolfberries (%).

#### 2.5.3. Preparation of Extract

The moisture content was 10 ± 0.5% of the dried wolfberry sample (1.0 g), which was macerated by mortar according to a solid–liquid ratio of 1:5 (m/V) in 50 mL of ethanol on ice to form a slurry and transferred to a triangular flask. The sample was centrifuged for 10 min (120 r/min) and placed for 48 h in a constant temperature shaker without light (4 °C, 4000 r/min). The supernatant (20 mL) was stored at 4 °C for the later determination of soluble sugar, total phenols, total flavonoids, and antioxidant capacity [38,39,40,41].

#### 2.5.4. Determination of Total Sugar

Wolfberry polysaccharide content was determined by the phenol sulfate method, as follows: A total of 5 μL of sample extract was added into a test tube with 1 mL of 9% phenol solution. Concentrated sulfuric acid (3 mL) was added after fully mixing, and the solution was allowed to incubate 30 min at room temperature. Absorbance at 485 nm was measured from three incubate aliquots and compared against a standard curve of polysaccharide content, with sucrose as the reference substance and without a sample solution as a blank control, calculated as follows [38]:(7)$$W=\frac{V_{2}C_{1}}{V_{1}M},$$
where *W* is the total sugar content of the wolfberries (g/g); *C*_1_ is the sucrose concentration (g/mL); *V*_1_ is the volume of the sample extract used in titration (mL); *V*_2_ represents the total volume of the sample extract (mL); and *M* is the quality of the dried wolfberries (g).

#### 2.5.5. Determination of Total Phenol Content

Total phenolic compounds in the wolfberries were determined by Folin’s phenol reagent test as follows: A total of 300 μL of sample extract was added to a test tube with 2 mL of 10% Folin–Ciocalteu reagent and 1 mL of 7.5% Na_2_CO_3_, then mixed and left to incubate for 60 min in dark conditions at 37 °C. Absorbance at 760 nm was measured from three incubate aliquots and compared against a standard curve of total phenolic content using gallic acid as the reference substance without a sample solution as a blank control, calculated as follows [39]:(8)$$P=\frac{V_{2}C_{2}}{V_{1}M},$$
where *P* is the total phenol content of samples (mg/g); *C*_2_ is the gallic acid mass concentration (mg/mL); *V*_1_ is the volume of titrated extract of the sample (mL); *V*_2_ is the total volume of the sample extract (mL); and *M* is the quality of the dried wolfberries (g).

#### 2.5.6. Determination of Total Flavonoid Content

Total flavonoids in the wolfberries were determined by the sodium nitrite-aluminum nitrate-sodium hydroxide method, as follows: A total of 1200 μL of sample extract was added to a test tube with 2 mL of distilled water and 0.3 mL of 5% Na_2_CO_3_ solution. A solution of 10% AlCl_3_ solution (0.3 mL) was added after mixed oscillation for 5 min, and 2 mL of 1 M NaOH solution was added after continued mixed oscillation for 1 min. Absorbance at 510 nm of three sample aliquots, taken after thorough mixing, was compared against a standard curve of total flavonoids content with catechin as the reference substance without a sample solution as a blank control, calculated as follows [40]:(9)$$F=\frac{V_{2}C_{3}}{V_{1}M},$$
where *F* is the total flavonoid content (mg/g); *C*_3_ is the mass concentration of catechin (mg/mL); *V*_1_ is the volume of titrated extract of the sample (mL); *V*_2_ is the total volume of the sample extract (mL); and *M* is the quality of the dried wolfberries (g).

#### 2.5.7. Determination of DPPH Radical Scavenging Activity

The antioxidant capacity of the wolfberries was determined by the DPPH method, detailed in the following steps. A total of 80 μL of sample extract was added to a test tube with 3 mL of 10^−4^ M DPPH methanol solution. Absorbance at 515 nm was measured from three incubate aliquots and compared against 70% ethanol as a blank control and 500 μM of 90% ascorbic acid as a positive control. The antioxidant capacity of the sample solution was expressed by the inhibition rate and calculated as follows [41]:(10)$$I=\frac{A_{0}-A}{A_{0}}\times 100,$$
where *I* is the inhibition rate of the sample solution (%); *A* is the absorbance of the sample solution; and *A*_0_ is the absorbance of the solution without the sample.

#### 2.5.8. Determination of V_C_ Content

Spectrophotometric determination of ascorbic acid in the wolfberries was performed as follows: The moisture content was 10 ± 0.5% of the dried wolfberry sample (1.0 g), which was macerated by mortar on ice in 20 mL of 50 g/L TCA solution to produce a slurry, then transferred to a 100 mL volumetric flask; the 50 g/L TCA solution was then added q.s. to a total volume of 100 mL. The supernatant was put on standby after being centrifuged at 4 °C and 1800 r/min for 10 min [42,43].

During the experiment, 1 mL of sample extract was taken and added to a test tube with 1 mL of 50 g/L TCA solution and 1 mL of absolute ethanol, mixed, and then 0.5 mL of 0.4% phosphoric acid-ethanol solution, 1 mL of 5 g/L red phenanthroline-ethanol solution, and 0.5 mL of 0.3 g/L FeCl_3_-ethanol solution were added by shaking until the mixture was homogeneous. Absorbance at 534 nm was measured from three solution aliquots and compared against a standard curve of V_C_ compound content with ascorbic acid as the reference substance without a sample solution as a blank control. The content of ascorbic acid in the wolfberries was expressed by the mass of ascorbic acid contained in a 100 g sample (fresh weight) and calculated as follows [42]:(11)$$A=\frac{V\times m^{\prime }}{V_{s}\times m\times 1000}\times 100\;\left(\mathrm{mg}/100g\right),$$
where *A* is the V_C_ of the sample (mg/100 g); *m*′ is the mass of ascorbic acid obtained from the standard curve (μg); *V* is the volume of titrated extract of the sample (mL); *V_s_* is the total volume of the sample extract (mL); and *m* is the quality of the dried wolfberries (g).

#### 2.5.9. Determination of Betaine Content

The content of betaine in the wolfberries was determined by high-performance liquid chromatography.

(1)Chromatographic conditions:Phase column: Merck RP-C_18_ (250 × 4.6 mm, 5 μm); mobile phase: acetonitrile-water (83:17, *v*/*v*); flow velocity: 1 mL/min; column temperature: 30 °C; detecting wavelength: 195 nm; injection volume: 1 μL.(2)Preparation of reference solution:Refined betaine (4 mg) was diluted and dissolved with methanol to prepare a 1 mg/mL reference stock solution. The reference solution of 500 μg/mL was obtained by adding 0.5 mL of methanol to 0.5 mL of reference solution.(3)Preparation of test samplesThe moisture content was 10 ± 0.5% of the dried wolfberry sample (1.0 g). which was placed in a porcelain mortar with 30 mL of methanol solution and macerated on ice to form a slurry, loaded in the triangular flask, and filtered after an ultrasonic treatment (power: 100 W; frequency: 40 kHz; time: 25 min).Filtrates were centrifuged in 10 mL centrifuge tubes, then supernatants were filtered through a 0.45 μm membrane. The subsequent filtrate was taken as the test sample. The peak area was recorded and the betaine content was calculated according to the external standard method; the calculation equation is as follows [43]:(12)$$B=\frac{C_{R}\times \left(A_{X}/A_{R}\right)\times D\times V}{m}\times 100\%,$$
where *B* is the sample betaine content (%); *C_R_* is the concentration of the reference substance (mg/mL); *A_X_* is the peak area of the sample; *A_R_* is the peak area of the reference substance; *D* is the dilution multiple of the test sample; *V* is the volume of the first preparation of the test sample (mL); and *m* is the injection volume of the test sample (mL).

#### 2.5.10. Microstructure Analysis

The moisture content was 10 ± 0.5% of the dried fixed-size wolfberry samples (2 × 2 mm), which were tested under different drying conditions, then post-fixed with 2.5% glutaraldehyde for 30 min to stabilize the microstructure prior to electron microscopy. Glutaraldehyde fixation was necessary because dried samples still contained a high moisture content. Gradient dehydration was performed with an ethanol solution series (50%, 70%, 80%, and 90%), followed by dehydration with anhydrous ethanol three times. Each dehydration time was 10 min. Final samples were immersed in 100% tert-butanol for 15 min and then placed in a vacuum freeze drying machine for processing [22,23,24,25,28,29,30]. The sample observation surface was placed upward and fixed on the scanning electron microscopy sample table with conductive tape after treatment, and the surface was coated with an ion sputter-coating instrument for 90 s, then placed under a scanning electron microscope. The acceleration voltage of the electron microscope was set to 5 kV and the magnification was set to 500×. Representative views were selected for microscopic photography.

#### 2.5.11. Statistical Analysis

The single-factor test method was adopted, each group was repeated three times, and all data were expressed as mean ± standard error (SE). Treatment comparisons were performed by a one-way analysis of variance (ANOVA) using SPSS 26.0 (IBM, Armonk, North Castle, NY, USA) with α = 0.05. Origin 2021 (OriginLab, Northampton, MA, USA) was used for mapping.

## 3. Results and Discussion

### 3.1. Analysis of Drying Characteristics

#### 3.1.1. Effect of Plate Spacing on Drying Characteristics

The effect of electrode spacing on the drying characteristics of wolfberries at a 0.025 MPa RF vacuum degree and 55°C hot air temperature is shown in Figure 2. The RF drying time decreased by 17.64% and 29.41% when the plate spacing decreased gradually from 100 mm and when the drying rate decreased from the peak to 0.27 g/h, 0.29 g/h, and 0.36 g/h after 8.5 h, 7 h, and 6 h, respectively. At 5 h, 7 h, and 6 h, the moisture ratio nodes reached 0.46 g/g, 0.47 g/g, and 0.47 g/g, respectively. The diagram indicates that the drying rate of wolfberries decreases with the increase in electrode spacing (Figure 2b), which is consistent with the variation of the RF load current with electrode spacing, where the temperature of the material increases rapidly at the beginning of RF drying, and a higher temperature is reached in a short time with the smaller plate spacing when the initial moisture content is constant. This was because the electric field intensity increased with the decrease in the plate spacing, more energy was absorbed, and the heating rate increased [22,23]. The residual unevaporated free water on the surface of the material was lost with the air during the transfer process from RF to hot air drying, which resulted in a temporary increase in the drying rate. RF heating makes the wolfberries have a positive temperature gradient from the center to the surface, causing a vapor pressure difference and improving the drying rate, and the material enters a low-amplitude stable heating stage after the rapid heating stage [28].

#### 3.1.2. Effect of RF Vacuum on Drying Characteristics

Figure 3 displays the effects of different vacuum pressures on the drying characteristics at an RF plate spacing of 90 mm and a hot air temperature of 55 °C. The moisture ratio node reached 0.42 g/g and the drying rate decreased from the peak to 0.15 g/h after 12 h of RF drying when the vacuum degree was 0.035 Mpa, prior to the transfer to the hot air drying stage. The RF drying time was shortened by 41.6% and 49.5% compared with that at 0.035 MPa when the vacuum degree gradually decreased to 0.025 MPa and 0.015 Mpa, respectively; the corresponding moisture ratio nodes were 0.47 g/g and 0.44 g/g, respectively. The observation showed that the total drying time decreased with the decrease in vacuum pressure, and the drying rate was negatively correlated with the vacuum pressure (Figure 3b). This is because the whole material is heated during RF drying. The surface and internal temperatures increase simultaneously, and the decrease in the vacuum causes the pressure inside the sample to exceed the external pressure and the surface moisture begins to evaporate, causing the drying rate to increase gradually [44,45]. However, the dielectric properties of wolfberry samples decreased with the decrease in moisture content, which reduced the RF energy absorbed by the samples, and the heat needed for the diffusion of bound water molecules was much higher than free water molecules. Therefore, the RF drying rate gradually decreased [18,19,46].

#### 3.1.3. Effect of Hot Air Temperature on Drying Characteristics

The effects of different hot air temperatures on the drying characteristics of wolfberries at an RF plate spacing of 90 mm and a vacuum degree of 0.025 MPa are shown in Figure 4. The hot air drying time was 14.5 h and the moisture ratio decreased to 0.12 g/g at the end of the drying when the temperature was 60 °C. The hot air drying time was shortened by 23.1% and 42.3%, in turn, and the moisture ratio was reduced to 0.11 g/g and 0.10 g/g when the temperature gradually increased to 55 °C and 60 °C, respectively. Figure 4b shows that the moisture ratio and drying rate decreased gradually during hot air drying, and the decreasing trend of the moisture ratio was more obvious at the higher temperature, with a faster average drying rate and less dehydration time required (*p* < 0.05). This is due to the heat-transfer resistance between the material and the external environment; thus, the surface temperature is much higher than the internal temperature, resulting in a large temperature difference between internal and external areas during hot air drying [20,47]. However, the temperature gradient inside and on the surface of the wolfberries was gradually reduced with the hot air drying, whereas the flow rate of water in the capillary and intercellular space accelerated, and the heat-transfer rate was improved [18,22].

### 3.2. Quality Analysis

#### 3.2.1. Color and Rehydration Rate

Table 1 compares the effects of natural drying, single hot air drying, and combined drying under different conditions on the color difference and rehydration rate of wolfberries. The results showed that the redness value *a** of natural-dried wolfberries was 43.93, which was higher than the fresh samples (*p* < 0.05). The epidermis brightness *L** was close to the fresh samples, and because the total color difference Δ*E* was smaller than in other drying conditions, it was possible that carotenoids in the wolfberries were degraded and accompanied by browning due to the influence of the climate and environment during drying. The epidermis brightness *L** and redness *a** values of dried wolfberries were significantly lower than fresh samples in the RF drying stage (*p* < 0.05). *L** values decreased from 33.84 and 36.06 to 31.01 and 32.16, respectively, and *a** values decreased from 30.15 and 31.59 to 28.02 and 26.95, respectively, when the plate spacing (from 80 mm to 100 mm) and vacuum degree (from 0.015 MPa to 0.035 MPa) gradually increased; however, the Δ*E* increased to 13.89 and 14.35. As the plate spacing and vacuum degree increased, the heating rate of the sample decreased and the environmental humidity in the RF drying differed from hot air drying, resulting in significant color difference changes [22,23,24,25]. The total color difference Δ*E* of dried wolfberries increased from 10.53 to 12.71 when the hot air temperature increased from 55 °C to 60 °C, which was significantly higher than natural drying (*p* < 0.05), but lower than single hot air drying (*p* < 0.05). This is due to the Maillard reaction of reducing sugars in wolfberries with free amino acids or with exposed amino acid residues on protein chains caused by the increase in temperature, resulting in the browning of the samples [22,35].

Table 1 shows that the rehydration capacity of the RF vacuum-dried wolfberries is significantly higher than the hot air-dried samples (*p* < 0.05), likely because the soluble solids accumulate on the surface of the sample during hot air drying, resulting in surface hardening and preventing internal moisture diffusion; furthermore, the crude protein, pectin, and other substances were seriously deformed, thus affecting the rehydration performance [35,47]. The overall heating and negative pressure environment of RF vacuum drying technology made the samples produce high vapor pressure, greatly reduced surface hardening, and caused the dried products to exhibit porous and loose microstructures. These characteristics improve rehydration performance [22,23].

#### 3.2.2. Effect on Total Soluble Sugar

As shown in Figure 5, the polysaccharide content (0.72 g/g) of the wolfberries which were dried by natural drying was the highest, whereas the other drying conditions significantly reduced the polysaccharide content (*p* < 0.05) because the heating rate of mechanized drying was fast and the Maillard reaction and caramelization of the sample were intensified, resulting in different degrees of sugar spillover. The polysaccharide content of the RF-hot-air-dried wolfberries was the highest (0.69 g/g) when the RF plate spacing was 100 mm, which was 20.3% higher than the content of the single hot air-dried ones (0.55 g/g). This is because the RF drying heats holistically, and the heating rate is slow when the plate spacing increases. Furthermore, the steady-state heating period of the material is longer and the heating uniformity is better [17,18,19], which effectively retains the sugar content. The polysaccharide content (0.62 g/g) at 0.025 MPa was 4.84% and 6.45% higher than at the other two vacuum degrees, respectively, and 12.7% higher than from single hot air drying. It is likely that the vacuum degree of 0.025 MPa can effectively avoid the glow discharge problem of wolfberries during RF drying, reduce the local overheating and burnt phenomenon caused by the uncontrolled effect of sample corner heating [44,45], and retain the sugar content in the sample. The polysaccharide content of wolfberries was 11.3% higher than from single hot air drying when the hot air temperature of combined drying was 55 °C (*p* < 0.05). This could have occurred because more cellulose in the sample at this temperature was degraded to soluble polysaccharides [35], thereby increasing the polysaccharide content.

#### 3.2.3. Effect on Total Phenol

As shown in Figure 6, the total phenol content of dried wolfberries by single hot air drying was 9.04 mg/g, which was 17.7% lower than the content (10.98 mg/g) of natural-dried berries (*p* < 0.05). This is because hot air drying has a shorter drying time and faster heating rate than natural drying. However, phenolic compounds have strong activity and unstable chemical properties; therefore, the higher temperature accelerates the oxidation and thermal degradation of phenolic compounds [48]. The total phenol content decreased by 10.2% and 15.9%, as compared with 100 mm plate spacing, with the decrease in the plate spacing during the combined segmented drying. This was primarily because the electric field intensity produced by RF increased with the decrease in electrode spacing [11,12], resulting in the destruction of wolfberry cell-wall polymers and the loss of more cell-wall phenolics or adhesive phenolics. The total phenol content was the highest (12.00 mg/g) when the vacuum degree was 0.035 MPa, and the content decreased by 16.6% and 24.35% when the vacuum degree was decreased. Due to the high boiling point of water under low vacuum conditions during the RF drying, the rapid evaporation of water altered the cell-wall structure and affected the activity of intracellular enzymes, which reduced the total phenol content. In the hot air drying stage, the total phenol content of dried wolfberries at 55 °C and 50 °C decreased by 17.5% and 19.3%, respectively, compared with those obtained at 60 °C (12.14 mg/g). This was because a longer drying time was needed at the lower temperature and polyphenol oxidase and peroxidase in the wolfberries increased the oxidation reaction time of phenolic compounds, resulting in a decrease in their content [49].

#### 3.2.4. Effect on Total Flavonoids

It can be seen from Figure 7 that the total flavonoid content of the dried wolfberries obtained by other drying methods was significantly lower than the content (2.84 mg/g) from natural drying (*p* < 0.05). A heat treatment can release flavonoids in the plant matrix, but a higher temperature causes polyphenol oxidase inactivation; therefore, the release of flavonoids decreased the decomposition reaction [50]. The total flavonoid content of the dried wolfberries increased first and then decreased under combined drying, which was significantly different from the content (2.04 mg/g) from single hot air drying (*p* < 0.05). The main reason may be that the synthesis of flavonoids is stimulated by dehydration stress during the drying process, and the later decline may be due to the decrease in water content, which makes wolfberries more vulnerable to oxidative stress [51]; thus, they consume a large amount of flavonoids. The total flavonoid content (2.60 mg/g) increased by 21.5% at the 90 mm plate spacing, 0.025 MPa vacuum pressure, and 55 °C hot air temperature as compared to the content from single hot air drying. It may be that the dehydration of the wolfberries under this condition induces the increase in the PAL enzyme activity, thereby increasing the content of total flavonoids [52]. The total flavonoid content of dried wolfberry products was the lowest (1.83 mg/g) when the hot air temperature decreased to 50 °C, which was 9.81% and 35.2% lower than the content from single hot air drying and natural drying, respectively. The appropriate temperature can cause cell-wall fragmentation without damaging flavonoids, which makes flavonoids easy to release and leads to an increase in total flavonoid content [50].

#### 3.2.5. Effect on Antioxidant Activity

The antioxidant activity of the dried wolfberries is shown in Figure 8 under different drying conditions. The main principle is to determine the scavenging capacity of the stable free radical 1,1-diphenyl-2-trinitrophenylhydrazine by the DPPH method [41]. This antioxidant capacity is expressed by the inhibition rate, and greater inhibition rates correlate with stronger antioxidant activity [41]. The antioxidant activity of dried wolfberries obtained by single hot air and combined drying methods was significantly enhanced (*p* < 0.05) compared to natural drying (I = 44.16%). The contact time of active antioxidants with air increases due to the long drying time of natural drying, and increased oxidation leads to the degradation of antioxidants [53]. The total antioxidant capacity of wolfberries after combined drying increased first and then decreased. The free radical scavenging rate of active substances was the highest under the conditions of a 90 mm plate spacing, 0.025 MPa vacuum pressure, and 55 °C hot air temperature, and the inhibition rate was 66.14%, which increased by 19.2% and 33.2% compared with natural drying and single hot air drying, respectively. It may be that during the alkali pretreatment under these parameters, RF-hot air combined drying enhanced the hydrogen atom and single electron supply capacity of phenolic compounds in the wolfberries, and the expression of antioxidant activity was more obvious [54].

#### 3.2.6. Effect on V_C_

Figure 9 shows that the V_C_ of the wolfberries dried by single hot air drying was 2.15 mg/100 g, which was 27.4% higher than from natural drying (*p* < 0.05) because hot air drying has a faster heating rate and is less time consuming than natural drying; therefore, the retention rate of V_C_ in rapid drying is much higher than that in slow drying. The loss of V_C_ also occurred in the combined segmented drying; however, the degradation of the dried wolfberries is still very low under the conditions of a plate spacing of 90 mm, vacuum pressure of 0.025 Mpa, and hot air temperature of 55 °C. The V_C_ was 3.18 mg/100 g, which was 32.4% higher than from single hot air drying (*p* < 0.05). This is because ascorbic acid is a sensitive substance, which is easily oxidized and degraded under a high temperature and aerobic environment [22,55]; therefore, the V_C_ of the dried wolfberries is significantly higher after the first stage of RF vacuum drying under a moderate temperature and low oxygen than from hot air drying.

#### 3.2.7. Effect on Betaine

Figure 10 shows that the content of betaine is the lowest (3.01%) after natural drying and the single hot air drying process increases it by 7.38% (*p* < 0.05); this is because betaine is an internal quaternary ammonium compound, similar to an amino acid amphoteric ion, and has a strong moisture absorption performance. Therefore, its content may be related to drying time and drying temperature, and since natural drying time is longer, it causes the large loss of betaine in wolfberries [56]. When the parameters of combined drying were a plate spacing of 100 mm, vacuum degree of 0.025 MPa, and hot air temperature of 55 °C, the betaine content of dried wolfberry products was up to 3.76%, which was 13.6% and 19.9% higher than in single hot air drying and natural drying processes, respectively. Because betaine is a metabolite, it is often used as an osmotic regulator to stimulate and accumulate under stress conditions, but its accumulation is closely related to the environment and betaine aldehyde dehydrogenase activity [57]. The pretreatment of the alkaline solution and the effect of an RF high-voltage electric field under appropriate parameters may be the dominant factors for the accumulation of betaine aldehyde dehydrogenase, thereby increasing the content of betaine in wolfberries.

#### 3.2.8. Microstructure Analysis

The microstructures of dried wolfberries under different drying methods are shown in Figure 11 and the macrostructures are shown in Figure 12. According to the microscopic structure diagram, the wax layer of the untreated fresh wolfberry skin is thicker, and the bundle strips are neatly and smoothly arranged closely with a large amount of wax debris attached (Figure 11a). The epidermis of the natural-dried wolfberries was arranged in a cord-like pattern, the cell tissue structure was clear, and there were some wax-film fragments (Figure 11b). The free water on the surface of the material cannot be evenly evaporated after natural drying, resulting in a local temperature rise on the surface of the wolfberries and burns, which significantly affect the sensory quality (Figure 12a). The surface layer of the wolfberries was loosely arranged in bundles after the alkali pretreatment and hot air drying, and there was an obvious gap between bundles. The local surface was damaged and concave, and the wax particles were basically removed (Figure 11c). Hot air convection drying can accelerate the drying rate and shorten the drying time of the wolfberries, but the alkali pretreatment destroys the wax layer. The introduction of a nonconnected and multidirectional heterogeneous porous network structure in the outer epidermis of the wolfberries makes the epidermal cells of the compact, internal tissue structure produce internal cracks [58], which lead to the aggravation of material wrinkles and affects the sensory quality (Figure 12b). Compared with other drying methods, the surface of the dried wolfberries was bundled and wrinkled after RF-hot air combined drying and the alkali pretreatment (Figure 11d). This may be due to the existence of pectin polymers in wolfberries in the form of long straight chains and large aggregates. The change of microstructure morphology during combined drying leads to the drying efficiency of dried wolfberries being improved [26]. However, the ionic wind and nonuniform electric field in the high-voltage electric field gradually degrade the long straight chain and large aggregates in the RF vacuum drying stage, and a large number of very small crystals appear on the surface of the wolfberries [59]. Moreover, due to the dielectric heating characteristics of the RF vacuum, water is forced to transfer from inside to outside, thus forming dense, porous drying products (Figure 12c).

## 4. Conclusions

In this study, fresh wolfberries were used as a raw material to study the effects of the RF-hot air combined segmented drying process on their drying characteristics, comprehensive quality, and microstructure. The results showed that with the decrease in plate spacing and vacuum pressure, and the increase in hot air temperature, the average drying rate of wolfberries increased and the drying time decreased. Based on the combined drying with the parameters of a plate spacing of 90 mm, vacuum degree of 0.025 MPa, and hot air temperature of 55 °C, the RF vacuum drying stage ended at 7 h, the moisture ratio node of transfer dry chamber was 0.47 g/g, and the drying rate decreased from the peak to 0.29 g/h. The total time to reach the safe moisture content (10 ± 0.5%) was 17 h, and the obtained dry products had a good color and a rehydration rate of 48.56%. We also noted that the combined drying under these parameters effectively retained the nutrients of the wolfberry. Among them, when the RF plate spacing was 90 mm, the antioxidant activity of the dried wolfberries was highest, and the inhibition rate was 66.14%, which increased by 19.2% and 33.2% when compared with the processes of natural drying and single hot air drying, respectively.

The total flavonoid content was highest (2.60 mg/g) when the vacuum degree was 0.025 MPa. The polysaccharide content (0.62 g/g) was 4.84% and 6.45% higher when the vacuum degree was 0.035 MPa and 0.15 Mpa, respectively. The total phenol content in dried products increased 12.7%. The V_C_ content of dried products was highest (3.18 mg/100 g) when the hot air drying temperature was 55 °C in the second stage. By comparing the microstructures of the wolfberry epidermides after different drying treatments, it was found that the local temperature on the surface increased after natural drying, which destroyed its flatness. The surface of dry products dried by hot air drying was rough and concave, and the wrinkles were aggravated. However, the surface of dried wolfberry products obtained by RF-hot air combined drying had bundle-like distribution and shrinkage aggregation, thus improving the drying rate, shortening the drying time, and improving the quality of dried products.

The study results showed that compared with traditional natural drying and single hot air drying, radio frequency-hot air combined drying improved the drying efficiency of the wolfberry, shortened its drying time, and significantly improved the quality and appearance, which is beneficial to improve the wolfberry commercial application status, enhance economic value, and provide a reference for the drying of similar materials.

## Figures and Tables

**Figure 1 foods-11-01645-f001:**
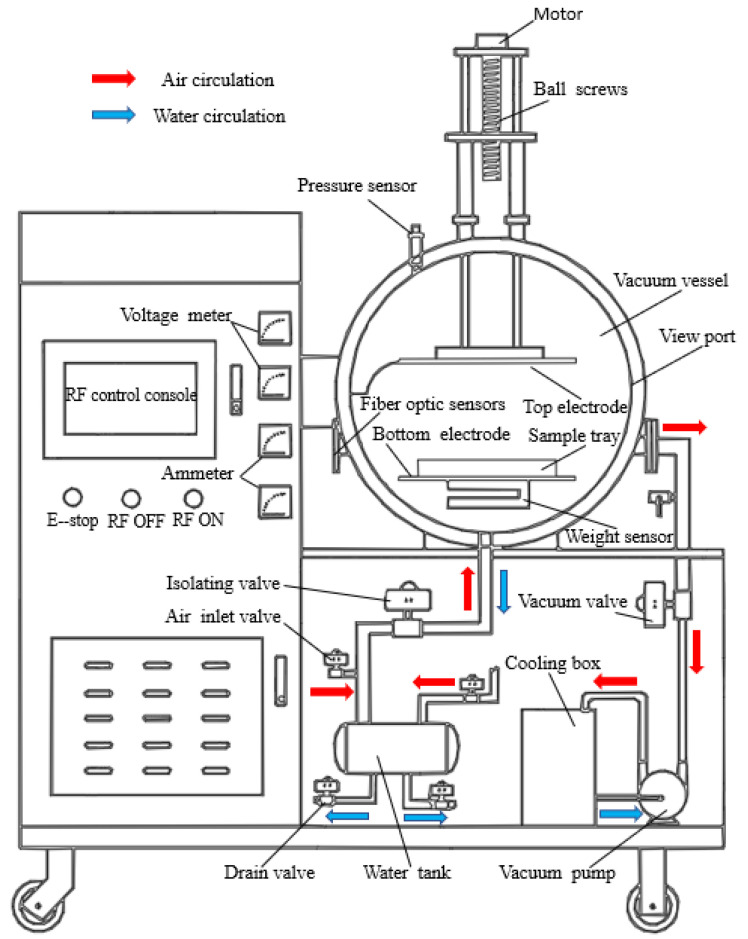
High-frequency vacuum medium-heating device.

**Figure 2 foods-11-01645-f002:**
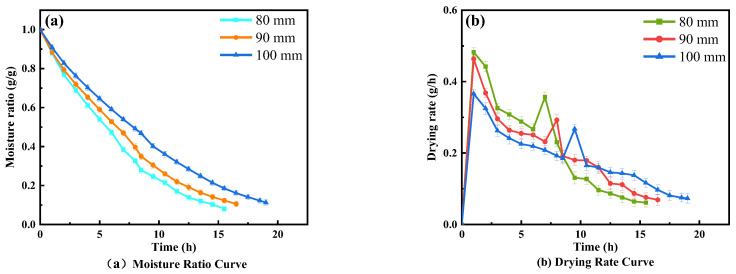
Effects of different plate spacing on drying characteristics.

**Figure 3 foods-11-01645-f003:**
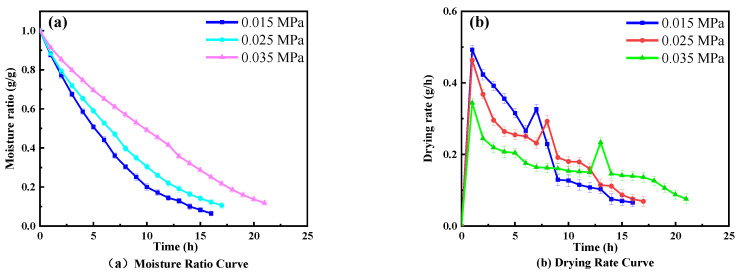
Effect of vacuum degree on combined drying characteristics.

**Figure 4 foods-11-01645-f004:**
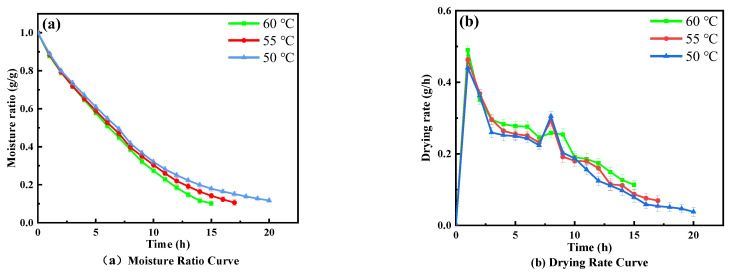
Effects of different temperatures on drying characteristics.

**Figure 5 foods-11-01645-f005:**
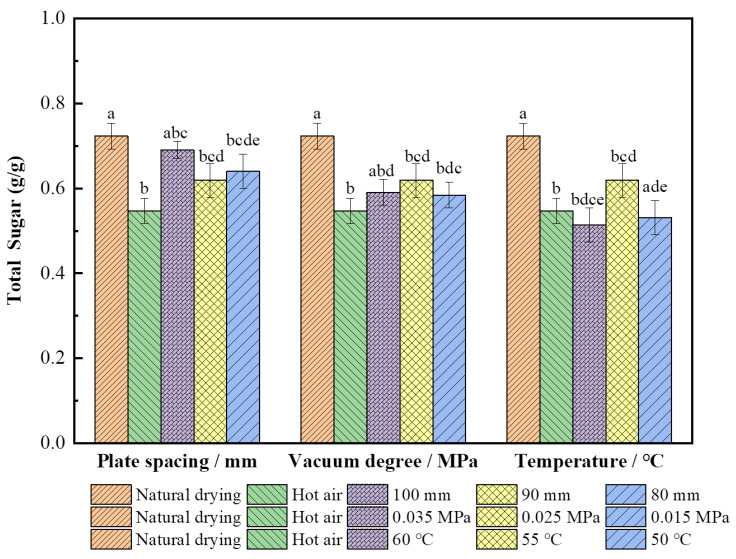
Total soluble sugar content of *L. barbarum* under different drying conditions. Note: Significant differences are represented by different letters (*p* < 0.05).

**Figure 6 foods-11-01645-f006:**
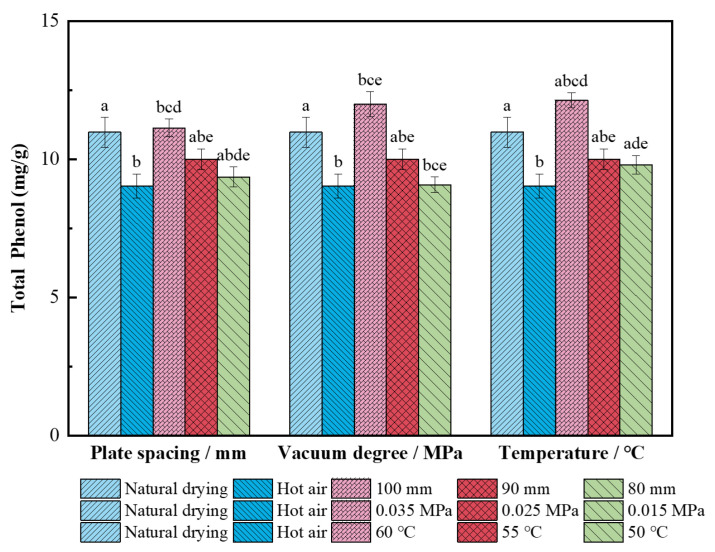
Total phenol content of *L. barbarum* under different drying conditions. Significant differences are represented by different letters (*p* < 0.05).

**Figure 7 foods-11-01645-f007:**
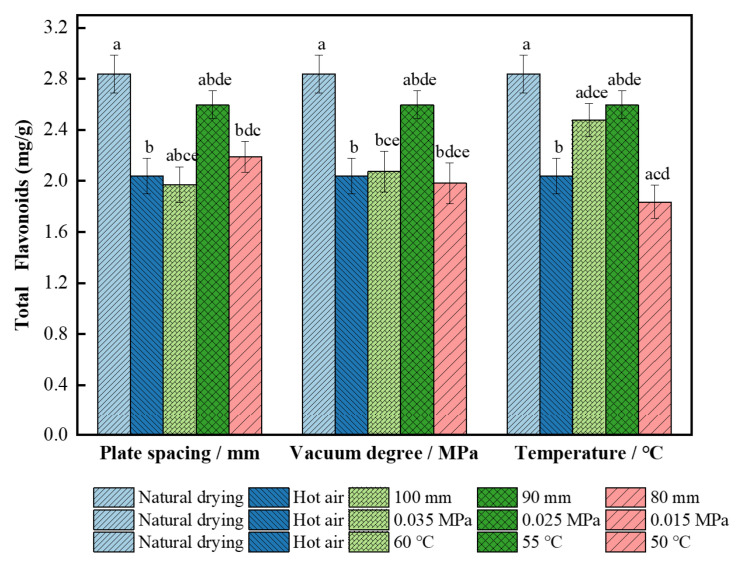
Total flavonoids content of *L. barbarum* under different drying conditions. Significant differences are represented by different letters (*p* < 0.05).

**Figure 8 foods-11-01645-f008:**
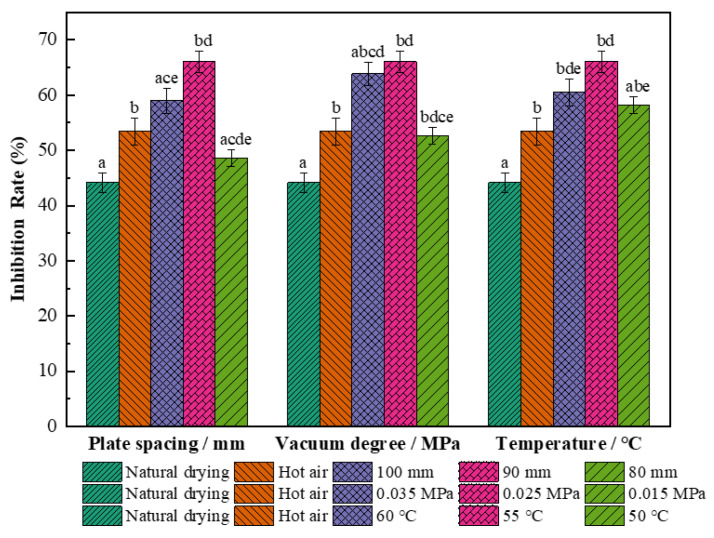
Antioxidant activity of *L. barbarum* under different drying conditions. Significant differences are represented by different letters (*p* < 0.05).

**Figure 9 foods-11-01645-f009:**
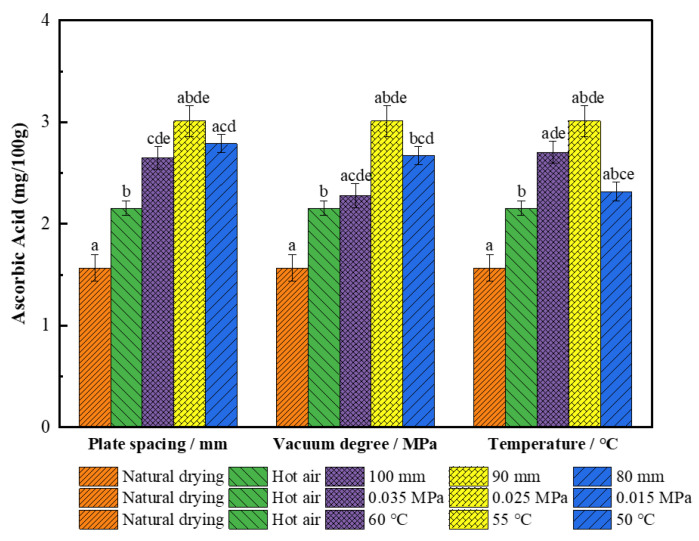
Ascorbic acid content of *L. barbarum* under different drying conditions. Significant differences are represented by different letters (*p* < 0.05).

**Figure 10 foods-11-01645-f010:**
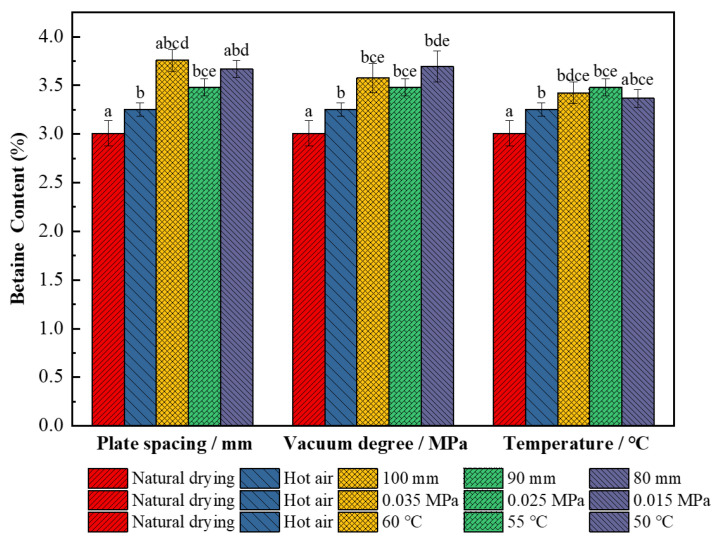
Betaine content of *L. barbarum* under different drying conditions. Significant differences are represented by different letters (*p* < 0.05).

**Figure 11 foods-11-01645-f011:**
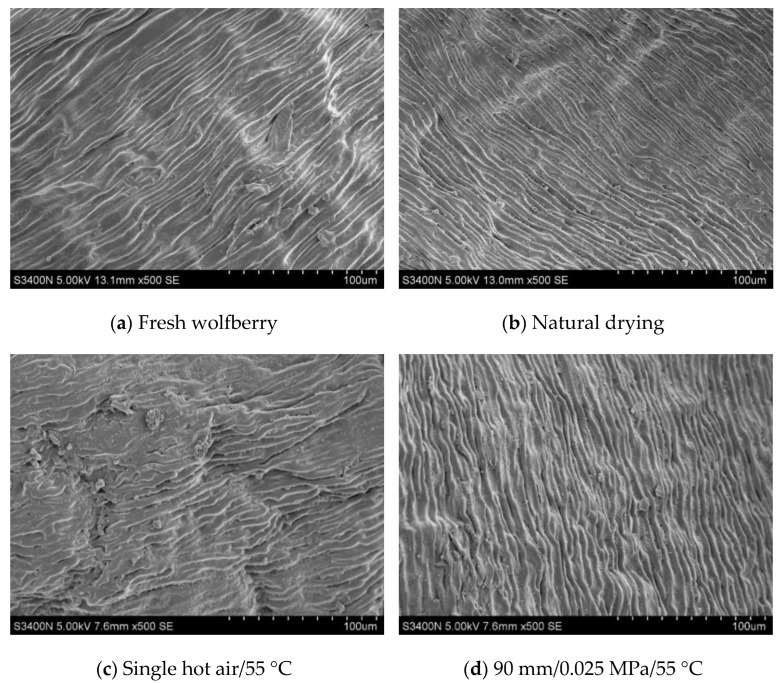
Microstructure of *L. barbarum* under different drying conditions.

**Figure 12 foods-11-01645-f012:**
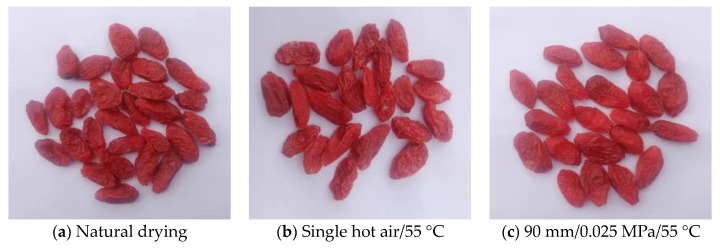
Macrostructure of *L. barbarum* under different drying conditions.

**Table 1 foods-11-01645-t001:** Color difference and rehydration rate of *L. barbarum* under different drying conditions.

Drying Conditions	*L**	*a* ^*^	*b* ^*^	ΔE	Rehydration Rate/%
Fresh Wolfberry	38.57 ± 0.81 ^ab^	39.39 ± 0.76 ^cd^	26.02 ± 0.93 ^ce^	−	−
Natural Drying	38.47 ± 0.38 ^ac^	43.93 ± 0.62 ^bde^	19.38 ± 0.59 ^ab^	8.56 ± 0.74 ^ac^	63.68 ± 0.53 ^be^
Single Hot Air/55 °C	34.15 ± 0.67 ^dc^	31.59 ± 0.74 ^ac^	15.69 ± 0.74 ^acd^	13.68 ± 0.57 ^bce^	35.82 ± 0.37 ^ce^
90 mm/0.025 MPa/60 °C	34.31 ± 0.72 ^bce^	32.34 ± 0.71 ^ce^	16.56 ± 0.72 ^bcd^	12.71 ± 0.59 ^cd^	43.85 ± 0.42 ^bcd^
90 mm/0.025 MPa/55 °C	33.54 ± 0.53 ^acd^	29.41 ± 0.66 ^bcd^	15.16 ± 1.37 ^cde^	11.48 ± 0.72 ^be^	48.56 ± 0.28 ^ab^
90 mm/0.025 MPa/50 °C	34.71 ± 0.96 ^cde^	30.88 ± 0.74 ^de^	16.30 ± 0.94 ^abe^	10.53 ± 1.02 ^ae^	50.57 ± 0.34 ^bcd^
100 mm/0.025 MPa/55 °C	31.01 ± 0.32 ^ae^	28.02 ± 0.98 ^ac^	18.37 ± 0.48 ^acd^	13.89 ± 0.94 ^cde^	41.89 ± 0.56 ^abe^
80 mm/0.025 MPa/55 °C	33.84 ± 0.57 ^cde^	30.15 ± 0.68 ^abe^	14.35 ± 0.26 ^ce^	10.21 ± 0.48 ^ac^	42.94 ± 0.29 ^acd^
90 mm/0.035 MPa/55 °C	32.16 ± 0.89 ^bde^	26.95 ± 0.22 ^bce^	13.76 ± 0.68 ^ac^	14.35 ± 0.26 ^abe^	37.15 ± 0.71 ^bce^
90 mm/0.015 MPa/55 °C	36.06 ± 0.86 ^bc^	31.59 ± 0.59 ^ae^	15.68 ± 0.27 ^ad^	11.29 ± 0.67 ^cd^	39.62 ± 0.52 ^ad^

Note: In the same column, the different lowercase letters reveal significant differences (*p* < 0.05).

## Data Availability

The authors confirm that the data supporting the findings of this study are available within the article.

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
