# Peer review of "Characteristics and Quality Analysis of Radio Frequency-Hot Air Combined Segmented Drying of Wolfberry (Lycium barbarum)"

_foods, 2022, doi:10.3390/foods11111645_

Round 1
Reviewer 1 Report
The article “Characteristics and quality analysis of radio frequency-hot air combined segmented drying of wolfberry (Lycium barbarum)” is well written, with proper methodology and results are well justified.
Abstract
Line 14: “parameters of plate spacing, vacuum degree, and hot air temperature”, mention the level of these parameters.
Provide data of drying rate and observed quality parameters in abstract.
Introduction
Line 30: “Solanaceae” is this a family of crop? If yes, mention as “family Solanaceae”.
Line 39, 40, 41: Compare critically about product qualities and energy consumption during dryig for the mention drying techniques.
Line 40-41: “energy utili-40 zation efficiency and product quality are difficult to control.” Explain the reason.
Line 51: Provide data of drying rate.
Recent studies on application of radio-frequency drying has been included but numerical data about the drying rate, drying temperature attained and data of percentage quality retention is not included. Author is suggested to include this information in the introduction.
The mechanism of radiofrequency drying is not written well. Author can include a pictography to present the drying mechanism.
Materials and Methods
In the introduction the need of this research is justified stating that it is difficult to control the energy efficiency of existing commonly used drying techniques. However no analysis is conducted to determine the energy efficiency of proposed drying technique.
Results and Discussion
Effect of Plate Spacing on Drying Characteristics: Authors have first explained general trend, the justified the reason for this trend and have given the numerical data at the end. Instead, numerical data should be explained before justifying the trend of results. Follow the same pattern throughout the manuscript.
Figure 2. Since there are two figures within figure 2, use Fig 2(a) & (b) in Figure 2 and within the text.
Line 294 to 299: The sentence is too lengthy and hence difficult to understand. Break the sentence appropriately.
Figure 3 & 4: Why drying rate starting from origin and is increasing at very high rate during early hours of drying? Explain the trend or change the format of the graph.
Line 350: What is natural drying and hot air drying? These drying methods are not explained in materials and methods.
Write about the energy efficiency in drying by radiofrequency drying technique.
Conclusion
In conclusion, only the summery of the results is repeated again. Write about how this drying method is more effective compared to existing techniques, the scope for commercial application and future research scope.
Author Response
Thank you again for your review and sugesstions on our manuscript。
Please see the attachment!

Reviewer 2 Report
The manuscript presents errors in the methods that bias some results. I allow myself to present some comments in summary but in an attached file I upload the manuscript with detailed comments (They are in yellow):
Introduction section:
The annual production of the fruit and consumption applications should be included, whether it is fresh or processed.
Methods Section:
-The geometric shape of the fruit is not indicated if it is whole or cut (the dimensions are missing).
-Missing information on color measurement
-Why was water at 80°C used? If the fruit is for fresh consumption, rehydration should be carried out at room temperature
-There is no clear design of the experiment.
-In the preparation of extracts for each treatment, the same humidity of the dry samples was not used, so that the results can be compared.
Results Section
-Some confusing analyzes and very long paragraphs are presented (see document)
--There is no analysis of variance (ANOVA) in most of the response variables.
-These results of quality parameters are not conclusive because they were not performed at the same food moisture level for each treatment.
-Procedures that are analyzed in the results, are not described in the methods, such as natural drying, application of Alkali treatment, among others.
Conclusion section
-They should be improved.
- Many results are presented.
It cannot be concluded that the pores increased and that it was denser because these were not measured. With the micrograph it is not possible to demonstrate exactly these parameters.
-
In an attached file I upload the manuscript with more detailed comments.
-

Author Response

(The authors gave the same response as above.)

Reviewer 3 Report
Retain of bioactive components and maintain/improve the sensory quality of food or food raw materials has high relevance for the food industry. Drying is a conventionally used food preservation technology, but it has a high-energy demand. The development of methods and processes to improve the dehydration and energy efficiency of drying can help to achieve better economy of food processing technologies and higher quality products. The manuscript has a special focus on the applicability of segmented RF/hot air drying for wolfberry investigating the drying kinetic, product colour, concentration of bioactive components and structural changes, respectively. Therefore, the topic of the manuscript is suit to the aims&scopes of the journal and can provide interesting information for the readers.
The manuscript is generally well written with a logic structure. Abstract summarizes well the main aims and results of the work. Introduction section provide information about the wolfberry, the general characteristics and applicability of RF drying and the relevance of combined RF/hot-air drying technology. The aims of the work and the specific research motivations are given well.
The parameters of drying equipments, experimental methods, drying kinetic calculation, analytical methods and microstructural analysis are given clearly and in details. The applied methods are adequate for the investigation of the parameters defined in specific research aims, and suit to the sample characteristics, respectively. The manuscript contains interesting results that are valuable not just for the science but also for the practice. Results are discussed with relevant references. Figures and tables represent well the experimental results.
Comments, suggestions:
Please give the raw material (wolfberry) as key word.
Please give and highlight the novelties of the research/manuscript (Introduction section).
It is not clear how was the operational parameter ranges selected/determined (temperature, pressure etc). Please give this information.
The effects of the different operational parameters (temperature, energy density, vacuum etc) on the energetic efficiency are not analysed. It should be given to select the best operational parameter ranges.
It should investigate the sensory properties of dried product, as well.
Please improve the visibility of Figure 2-5 (axis title etc).
Please unify the reference style.
Author Response

(The authors gave the same response as above.)

Round 2
Reviewer 1 Report
I am happy with the revision and thanks the authors for taking the comments for revision.
Author Response
Thank you.
Reviewer 2 Report
Dear editors
The corrections I consider to be fine, however the authors did not take into account additional comments that are in the manuscript file that I uploaded to the system. It is important that they review and correct.
